# The Good, the Bad, and the Useable Microbes within the Common Alder (*Alnus glutinosa*) Microbiome—Potential Bio-Agents to Combat Alder Dieback

**DOI:** 10.3390/microorganisms11092187

**Published:** 2023-08-30

**Authors:** Emma Fuller, Kieran J. Germaine, Dheeraj Singh Rathore

**Affiliations:** 1EnviroCore, Dargan Research Centre, Department of Applied Science, South East Technological University, Kilkenny Road, R93 V960 Carlow, Ireland; emma.fuller@teagasc.ie (E.F.); kieran.germaine@setu.ie (K.J.G.); 2Teagasc, Forestry Development Department, Oak Park Research Centre, R93 XE12 Carlow, Ireland

**Keywords:** biocontrol agents, *Phytophthora*, microbiome, beneficial microbes, pathogens, PGPR, PGPF

## Abstract

Common Alder (*Alnus glutinosa* (L.) Gaertn.) is a tree species native to Ireland and Europe with high economic and ecological importance. The presence of Alder has many benefits including the ability to adapt to multiple climate types, as well as aiding in ecosystem restoration due to its colonization capabilities within disturbed soils. However, Alder is susceptible to infection of the root rot pathogen *Phytophthora alni*, amongst other pathogens associated with this tree species. *P. alni* has become an issue within the forestry sector as it continues to spread across Europe, infecting Alder plantations, thus affecting their growth and survival and altering ecosystem dynamics. Beneficial microbiota and biocontrol agents play a crucial role in maintaining the health and resilience of plants. Studies have shown that beneficial microbes promote plant growth as well as aid in the protection against pathogens and abiotic stress. Understanding the interactions between *A. glutinosa* and its microbiota, both beneficial and pathogenic, is essential for developing integrated management strategies to mitigate the impact of *P. alni* and maintain the health of Alder trees. This review is focused on collating the relevant literature associated with Alder, current threats to the species, what is known about its microbial composition, and Common Alder–microbe interactions that have been observed worldwide to date. It also summarizes the beneficial fungi, bacteria, and biocontrol agents, underpinning genetic mechanisms and secondary metabolites identified within the forestry sector in relation to the Alder tree species. In addition, biocontrol mechanisms and microbiome-assisted breeding as well as gaps within research that require further attention are discussed.

## 1. Introduction

Over the last two decades, there has been an elevated interest regarding the significance of native tree species, such as Common Alder (*Alnus glutinosa* (L.) *Gaertn*) [1], also known as Black Alder, European Alder, or just Alder. The genus *Alnus* belongs to the family *Betulaceae*, the birch clade [2]. Alder is known to be a short-lived, light-demanding, and water-demanding species [1]. In addition, Alder preferentially grows in non-shaded areas with little competition; however, it has the ability to naturally self-prune, as branches tend to die with lack of light [3,4]. This species typically contains short stalked reddish-brown winter buds, small flat waxy winged seeds, and woody cone-shaped fruit (catkins) [4,5]. There are more than thirty *Alnus* species worldwide, including shrubs and trees, with *A. glutinosa* being the primary native species within Europe [6,7]. While some other well-known Alder species include Italian Alder (*A. cordata*), Grey Alder (*A. incana*), Red Alder (*A. rubra*), White Alder (*A. rhombifolia*), Japanese Alder (*A. japonica*), and American Green Alder (*A. viridis*) [1,5]. Alder has demonstrated phenomenal resilience due to its adaptation to a variety of climates around the world [8]. This resilient species tolerates frost and survives in an extensive range of temperatures [9]. *A. glutinosa* has been found on a broad range of sites; however, it favours wet, riverine, and fertile areas with high humidity [3,10]. Preferable growing sites include riverbanks, ponds, surrounding lakes, streams, marshy waterlogged areas, shaded mountainous regions, wet woodlands, wet soils, and highlands with sufficient moisture content [10,11]. This species tends not to grow well on calcareous soils, acidic peat, arid sandy soils, and areas with stagnant water [1,12]. Alder has been found in and/or introduced to regions of Africa, Australia, Canada, India, Japan, Russia, North America, South America, and New Zealand [5,12,13,14].

Historically, Alder was considered to generate an inferior class of wood with low economic value [15]. A black dye, known as ‘poor man’s dye’, was produced from Alder bark and catkins and generally used for tanning leather [15]. Alder had an association with war and was known as the ‘tree of death’ since when cut, the light-coloured wood swiftly oxidises to a vivid red, giving a ‘bleeding’ effect [15,16]. Alder wood was used for items such as war shields, wheels, bowls, tubs, and troughs [15]. Alder has been used within many regions worldwide for various traditional health and healing medicinal practices, due to the presence of chemical constituents and biological components such as flavonoids, terpenes, phenols, saponins, and steroids [17]. For example, Common Alder bark has been shown to have medicinal benefits and has been used for treating burns, diseases, and infections, as well as potential antitumor activity [17]. Acero et al. (2012) conducted an ethno-pharmacological study of Common Alder bark and observed that it potentially has anti-oxidant and anti-inflammatory properties [18]. Fresh Alder catkins have shown antioxidant, antimicrobial, and anti-inflammatory properties due to the presence of polyphenols and certain microorganism strains [19].

Alder tends to have an extensive rooting system and forms large nitrogen-fixing root nodules via symbiosis with nitrogen-fixing filamentous bacteria, such as the genus *Frankia* [20]. Thus, this actinorhizal species has the pioneering ability to improve soil conditions by increasing organic matter and nitrogen content within soils to improve fertility, contributing to biodiversity so other plant populations can grow in the area. Furthermore, Alder can colonise areas that have previously been subject to significant disturbance and/or degradation, playing a vital role in ecosystem and land restoration [1,21]. Other ecological benefits of Alder roots include the reduction of flooding and soil erosion, water filtration/purification, and riverbank stabilisation [4,5]. Alder is a biodiverse habitat providing shelter for numerous plants, animals, and microbes. For example, otters and fish tend to use Alder roots surrounding waterbodies for nesting purposes as well as shelter to reduce predation [11]. Alder provides an important food source for several birds, fungi, lichens, mosses, and approximately 140 insects and mites, resulting in an additional food source for fish if these insects feed on trees planted beside waterbodies [3,22]. For example, Alder leaves provide a food source for plant-eating invertebrates and Alder catkins provide a source of pollen for bees as well as nectar and seeds for many bird species. The nitrogen-fixing capabilities of Alder allow for the production of nitrogen-rich leaf litter due to the presence of invertebrate detritivores and microbial decomposition, which provides a primary food source for invertebrates that are later eaten by bigger organisms, creating multiple food chains [23].

Its porous wood has various uses for carpentry, building, furniture production, biomass production, instrument manufacturing, cabinetry, dyes, and manufacturing charcoal [3,5,16]. Its timber has high durability and decay resistance beneath water, so it is frequently used for bridge piles, small boats, water structure supports, and jetties [4,24]. The climatic adaptation of *A. glutinosa*, coupled with the worldwide distribution of this species has substantially increased its importance and promising utilisation [8]. As a result, Alder breeding programs have been established; therefore, greater amounts of *A. glutinosa* are being cultivated within commercial forests in order to generate trees to produce timber [10,16]. As part of the European Union (EU) forest strategy for 2030, policies on broadleaf and biodiversity have been put in place to improve the quantity and quality of European forestry; thus, since Alder is a broadleaved species, the volumes of Alder plantations throughout Europe has increased [25].

There are numerous existing threats to Alder cultivation, some of which are abiotic such as drought [26,27], whilst others can be biological such as pathogenic infections [28]. Research regarding the relationship between pathogenic microbes and native trees has increased because of the elevated levels of disease spread throughout the world, which has caused damage to and the depletion of numerous tree species. Thus, a greater understanding of *Alnus*-associated beneficial microbes could provide an ecological, environmentally sustainable solution to the biological control of *Phytophthora* infection. Therefore, to gain a better understanding of the beneficial and pathogenic microbiota associated with *A. glutinosa*, this review reports the known microbiome of Alder, pathogenic threats of Alder, as well as any biological solutions available to date for the control of these pathogenic threats.

## 2. The Microbiome of Alder Species

Microorganisms are extremely diverse and plentiful within the environment. The type and abundance of microbial communities that exist within an organism are influenced by the environmental conditions surrounding plant communities as well as soil type, indicating the presence of selective communities associated with different plant species. Forest trees exist in close association with a diverse range of microbial organisms that play a crucial role in maintaining tree health, nutrient conditions, and ecosystem functions. This association can be mutualistic, parasitic, or symbiotic. Combined, these microbial communities associated with the tree are known as its microbiome. The composition of the microbial community can vary due to a range of factors including climate, edaphic conditions, anthropogenic activities, silviculture management practices, and various other events and stressors [29]. Organic matter and associated decomposition products also cause changes within forest soils, causing soil physicochemical alterations, acidification, and supply/leaching of nutrients [30]. The long-lived nature of trees provides a secure food source for beneficial microbiota as well as parasites and pathogens. In comparison to annual crops, underground microbial communities related to trees may be more consistent and have stable interactions due to the deep rooting system, and a constant energy flow system ranging from photosynthesates being pumped into the soil to the abscission of leaf/flower and fruit material, leading to organic matter build up in the soil and the absence of soil disturbance [29].

Tree roots and rhizospheric soil tend to have a particular and plentiful microbial community due to the occurrence of nutrient and mineral exchange between the soil microbiota and tree-component microbiota present in root tissues, ectomycorrhizal fungal roots, arbuscular mycorrhiza roots, and fungal mycelia [31]. These roots have the ability to alter the composition of the microbial community within the soil due to the compounds and root cells that are released into the soil [29]. Throughout tree development, some soil physiochemical properties can change, which causes alterations to the microbial communities present in the rhizosphere that modify tree morphology, promote growth, and enhance nutrient and mineral content [32]. Furthermore, beneficial and/or pathogenic bacterial and fungal communities are also present within the endosphere and phyllosphere microbiota of forest trees, which create ecological interactions such as mutualism, commensalism, and/or antagonism [33]. Beneficial microbiota include plant growth-promoting rhizobacteria (PGPR), plant growth-promoting fungi (PGPF), and biocontrol agents that have the ability to economically and efficiently alter the metabolism of tree substances in order to improve the growth, performance, and resistive properties of the species. Moreover, beneficial microbiota are necessary for the natural degradation of plant residues, since greater amounts of microbial assemblages are present within residues, which additionally degrade them into soil organic matter containing beneficial macromolecules and micromolecules. This process provides increased soil protection and sustains nutrient capacity. Furthermore, when plants develop under stressful environmental conditions, the plant-associated microbiota tends to increase levels of hormones such as abscisic acid and jasmonic acid, which help plants regulate growth and adapt to extreme conditions [33]. Plant growth-promoting microbes tend to be found on the root surface or as endophytes within plant tissues, which have the ability to act as biocontrol agents, bio-fertilisers, and/or bio-stimulants [34]. Due to the complex relationships and interactions within forest ecosystems, there are still many unknowns and potential beneficial microbes associated with the microbiome of trees; therefore, it is important that research on this topic continues.

Plant growth-promoting rhizobacteria (PGPR) play a significant role in the forest ecosystem by releasing numerous regulatory molecules, aiding the growth and development of forest trees [35]. Deciduous forest soils tend to contain a wide array of bacterial phyla, which include Acidobacteria, Actinobacteria, Proteobacteria, Bacteroidetes, and Firmicutes [36,37]. PGPR can be found in the rhizosphere, phyllosphere, and/or endosphere and have the ability to promote plant growth in different ways including nitrogen fixation, the production of the plant hormones such as auxins (e.g., indole acetic acid (IAA)), enzymes (1-aminocyclopropane-1-carboxylic acid (ACC) deaminase, cellulase, chitinases, etc.) and siderophores, as well as the ability to compete with pathogenic microbes to minimise the spread of infection [33]. IAA is a plant hormone that regulates growth and development [38]. ACC deaminase is produced in the presence of ACC, the immediate precursor to the plant stress hormone ethylene, when plants grow within stressful conditions [39]. ACC deaminase aids plant survival by decreasing the amount of ethylene present, as excessive amounts of this plant hormone can have negative effects on plant growth [40]. Siderophores have the ability to uptake iron from the rhizosphere to promote plant growth and contribute to nutrition whilst minimising the amount of iron available for harmful pathogens [41]. Cellulase has been known to promote the breakdown of fungal and plant cell walls, aid soil fertility by speeding up the decomposition of plant residues, reduce spore germination, and reduce fungal growth [42,43,44]. Richter et al. (2010) analysed cellulase activity to suppress *P. cinnamomi*, a root rot pathogen of avocados and reported that sporangia production was reduced when cellulase enzymes were present, indicating that cellulase enzymes may have the ability to decelerate the spread of *P. cinnamomi* within avocado sites [45]. Pectinase is an essential enzyme for plant growth and development as it breaks down pectin within cell walls, allowing new plant growth to occur as well as providing entry for beneficial microbes to live endophytically [46]. Other mechanisms that PGPR induce in order to promote the health and growth of plants include the production of exopolysaccaride biofilms, organic acids to increase phosphate solubilisation, hydrogen cyanide to help with biotic stressors, and the production of cytokines and gibberellins which aid growth and development [35]. In addition, some PGPR can reduce oxidative stress by producing abscisic acid as well as stimulating growth by altering physiological mechanisms using emitted bacterial volatile organic components [35]. Some PGPR studies in particular have focused on analysing the interaction between the *Betulaceae* family and identified beneficial microbes including Actinobacteria genera (*Frankia*, *Streptomyces*) [47,48,49,50], an endophytic bacterial genus (*Rhizobium*) [51], *Bacillus* isolates (*B. licheniformis*, *B. pumilus*, *B. megaterium*, and *B. longisporus*) [52,53,54], and Gammaproteobacteria (*Pseudomonas*) [50,55]. Other known PGPR particularly in agriculture include *Arthrobacter*, *Azospirillum*, *Azorhizobium*, *Burkholderia*, *Flavobacterium*, *Mesorhizob*, and *Methylobacterium* [56]. With the evident advantages of these microbes in agriculture, there is great potential to exploit these PGPR with further research in the forestry sector.

The inoculation of plants with plant growth-promoting fungi (PGPF) has demonstrated many health benefits including enhanced seed germination, seed vigour, root morphogenesis, pathogenic suppression, the reduction of (a)biotic stressors, as well as an improved process of photosynthesis and mineralization [57]. The roots from the *Alnus* species have a tendency to form stable long-term biological interactions with mycorrhizae, including ectomycorrhizal fungi and/or arbuscular mycorrhizal fungi [58]. This symbiosis helps to increase the uptake of water and nutrients through the roots, improve tree health, growth, and reproductive ability, as well as provide a degree of tolerance to (a)biotic stressors [58]. For example, Thiem et al. (2020) showed that the inoculation of Alder seedlings with an ectomycorrhizal fungus, *Paxillus involutus* OW-5, promoted growth and increased tolerance in saline soils [59]. Culturable dark septate endophytes (DSEs) have the potential to promote plant growth particularly in metal-contaminated soils due to their capacity to provide a degree of phytoremediation/phytostabilization of heavy metals, increasing nutrients within soils, and produce the IAA hormone [2,60]. Fungal melanin tends to be found alongside some PGPF, which may help promote plant growth by providing a higher tolerance/resistance to environmental stress and promoting colonisation [58,61]. Fungal melanin exhibits beneficial biological functions including photo-protection, metal binding, mechanical protection, energy harvesting, cell development, antioxidant functions, anti-desiccant functions, chemical protection, as well as thermoregulation [61]. Some examples of identified PGPF that interact with the *Betulaceae* family include arbuscular mycorrhizal fungi (*Gigaspora rosea*), ectomycorrhizal fungi (*Hebeloma* sp., *Helotiales* sp., *Geopora* sp., *Thelephora* sp., *Tomentella* spp., *Paxillus involutus*, *Tylospora*, *Leccinum*, and *Rhizopogon*), endophytic fungi (*Cryptosporiopsis* spp., *Rhizoscyphus* spp.), DSE (*Phialocephala*), and ericoid fungi (*Oidiodendron*) [2,29,47,58,62,63,64,65]. In addition to all the above-mentioned PGPF, other known growth-promoting microbes used in agriculture belong to the genera *Alternaria*, *Chaetomium*, *Penicillum*, *Phoma*, *Serendipita*, and *Trichoderma* [66]. There has been a major growth of interest regarding PGPF acting as bio-fertilisers in agriculture due to their ecological benefits and plant improvement. In particular, the genera *Trichoderma*, *Penicillum*, and *Aspergillus* are the most studied within agriculture as they can promote crop growth and act as eco-friendly biological control agents to promote resistance/tolerance to biotic stresses [67]. These fungal genera may have potential plant growth-promoting properties within forestry, if not already used within this sector. Furthermore, exploring such unknown genera in forest tree species and soils would lead to novel microbes that can benefit both forestry and agriculture for sustainable production.

### What Is Known about the Microbiome of Alder?

To date, there have been few studies regarding microbes associated with *A. glutinosa*. Much of the information that is available focuses on the microbial communities present within *Alnus* roots and their associated rhizosphere. These studies particularly focus on soil salinity [68], inoculation with nitrogen-fixing bacteria [69], colonisation capabilities [70], plant growth promotion [30], antagonistic effects against pathogenic microbes [71], as well as adaptation to extreme environmental conditions [72]. Alder-associated bacterial microbes that have been identified belong to the phyla Actinobacteria (*Frankia alni*), Acidobacteria (*Granulicella*), Alphaproteobacteria (*Caulobacteraceae*, *Rhizobiales*, and *Sphingomonas*), Bacteroidetes (*Chitinophagaceae*, *Flavobacteriaceae*, and *Sphingobacteriaceae*,), Betaproteobacteria, Firmicutes (*Bacillus licheniformis* and *Bacillus pumilus*), Gammaproteobacteria, Pseudomonadota (*Bradyrhizobium*, *Oxalobacteraceae*, *Pseudomonas*, *Rhizobium*, *Rhodanobacter*, and *Xanthomoadaceae*), and Thermoleophilia [20,29,30,36,47,48,50,52,53,65,73]. Phyla of fungal microbes associated with Alder species include Ascomycota (*Cryptosporiopsis*), Basidiomycota (*Tomentella*, *Thelephora*), Ectomycorrhizal fungi (*Alnicola*, *Lactarius*, and *Phialocephala*), Glomeromycota (*Gigaspora rosea*), and Zygomycota (*Rhizocyphus*) [2,20,47,58,60,65,74]. See Figure 1 for a summary of the microbes identified in the different compartments of the *Alnus* species. With the economic and environmental importance of Alder coupled with its pathogenic threat, research regarding the root and rhizospheric microbiomes may potentially grow in the future.

Very little is known about the leaf, bark, and catkin microbiomes of Alder trees. A greater focus has been placed on analysing the chemical and biological components, as well as the secondary metabolites present in the bark, leaf, and fruit due to the medicinal value of these components [39,40,41]. Sukhikh et al. (2022) examined the antioxidant properties of *A. glutinosa* female catkins and found high levels of methanol extracts, ellagic acid, and ethyl acetate indicating their potential use as natural antioxidants [75]. Thiem et al. (2020) analysed the correlation between salt stress and mycorrhizal fungi from *A. glutinosa* growing in saline soil. Three fungal species were isolated from *A. glutinosa* catkins, which were *Amanita muscaria*, *Paxillus involutus*, and *Gymnopus*. Seedlings were later inoculated with each fungus to determine their effects against salt stress. It was observed that *P. involutus* aided in promoting the growth of seedlings and showed a degree of salt tolerance [59]. There have been studies on the metabolites and medicinal extracts of Alder bark, but no studies were found for the analysis of Alder tree bark microbiomes. Alder leaves have been studied regarding their medicinal potential. For example, Mushkina (2021) investigated the ability of *A. glutinosa* leaf tinctures to be used as wound-healing gels. It was determined that this gel had the ability to regenerate and heal wounds due to the presence of flavonoids, tannins, and phenolic acid [76]. Leaf extracts have been used to analyse their antagonistic effects against pathogens such as *Staphylococcus aureus*, *Bacillus subtilis*, *Escherichia coli*, *Pseudomonas aeruginosa*, and *Candida albicans* [77]. Concerning the microbial community within Alder leaves, there are few studies regarding this topic. Kayini and Pandey (2010) isolated fungal microbes from Nepalese Alder leaves including *Alternaria alternata*, *A. raphani*, *Aspergillus niger*, *Cladosporium cladosporioides*, *Epicoccum purpurascens*, *Fusarium oxysporum*, *Gliocladium roseum*, *Mucor hiemalis*, and *Pestalotiopsis* sp. [78]. A greater focus is needed regarding the microbial communities present within Alder leaves, bark, and catkins.

## 3. Microbial Pathogens of Alder Species

There are several pests and microbial pathogens associated with Alder. For example, Alder yellows phytoplasma disease (caused by a phytoplasma bacterial parasite) has been identified within numerous European countries, which causes stunted growth, yellowing of leaves, reduction of leave size and amount, as well as dieback and/or death [79,80,81,82]. The Ascomycota fungus, *Taphrina alni*, is a causal agent of Alder tongue galls on female catkins and has been identified throughout Europe. These galls are known as Alder tongues, as they are green-red elongated structures (depending on the season) [83]. The fungus *Septoria alnifolia* has been known to form leaf spots, stem cankers, and stem breakage on Alder trees [84]. Furthermore, a rust fungus, *Melampsoridium hiratsukanum*, causing yellow–brown spotting on Alder leaves, followed by early leaf fall, crown thinning, and/or death, has been identified [85]. The fungal pathogen *Mycopappus alni* tends to cause brown blotches on leaves and defoliation of Alder trees [86]. The bacterium *Erwinia alni*, has been found to cause bark cankers and bleeding and eventually kill off branches and/or the tree as a whole if the infection is severe [87]. The bacterium *Pseudomonas syringae* has been reported to cause leaf necrosis and the dieback of Common Alder [88]. With regards to insects, for example, the Alder leaf beetle, *Agelastica alni*, has been identified in Europe causing damage and defoliation of Common Alder [89]. Several species of Alder sawflies have been identified feeding on Common Alder such as *Monsoma pulveratum* [90], *Eriocampa ovata* [91], *Hernichroa crocea*, and *Cimbex conatus* [92]. A leaf miner, *Fenusa dohrnii*, was identified as a pest of Alder that causes damage [93]. There is an extensive list of pests and pathogens associated with Alder; however, not all tend to have damaging effects on this genus. For example, Sims (2014) discusses different types of fungal pathogens and insects that have been identified as associated with red, white, and thin-leaf Alder in western Oregon [94]. McVean (1953) extensively lists the insects and mites that associate with Common Alder as a food source [9], however not all of them are of economic importance and do not pose a severe threat to Alder trees. One of the major pathogens of Alder trees is the plant pathogenic oomycete species *Phytophthora*.

The species *Phytophthora*, meaning ‘plant destroyer’, has been associated with the dieback and decline of Alder across Europe [95]. There are approximately 200 identified and accepted species of *Phytophthora*, with more species unnamed/unidentified and likely to be discovered [96]. The pathogen typically infects the tree roots, causing them to rot, and spreads throughout the tree to cause damaging effects such as crown rot [97]. Infection can be caused via *Phytophthora* chlamydospores, hyphae, oospores, sporangium, and more frequently zoospores. Flagellated *Phytophthora* zoospores have a similar morphology to filamentous fungi but mainly have diploid hyphae and a cell wall comprising cellulose and/or β-glucans [97]. *Phytophthora* show phylogenetic similarities to algae and diatoms due to the structural composition of zoospores as well as the release of similar sexual antheridia (male) and oogonia (female) [96,98]. *Phytophthora* are soil-borne water moulds that have a strong dependency on water and humidity; therefore, the zoospores typically disperse and spread throughout water systems, eventually infecting tree roots [96,99]. In terms of the pathogen’s life cycle, sporangia tend to form when chlamydospores (asexually) and/or oospores (sexually) are germinated within wet environments which results in zoospores being released from the mature sporangium [97,100]. Oospores are formed when oogonia are fertilised by antheridia of the *Phytophthora* species. Zoospores swim through water within wet soils where they have the ability to form cysts on susceptible tree roots and bark near the root collar [100]. These cysts germinate, forming mycelia, which allows the pathogen to grow and spread biotrophically throughout the plant tissues where reproduction occurs and the life cycle begins again by producing chlamydospores, oospores, and sporangium via germination within plant tissues, spreading from the roots and further throughout the tree (Figure 2) [97,100]. Infection via tree roots results in damage which causes a reduction in water and nutrient uptake, impairing their availability for the rest of the tree compartments. The lack of minerals causes harmful effects such as leaf stomata closure and a reduction in photosynthesis [101].

Nave et al. (2021) inoculated Alder roots with *Phytophthora alni*, and after three weeks, oogonia, oospore, sporangium, antheridium, and mycelium were found at different stages of the reproductive cycle throughout the plant tissues [103]. *P. alni* hyphae has the ability to grow from Alder roots into the bark via secondary growth, travelling throughout the tree and infecting the cambium layer and adjacent phloem and xylem by continuously reproducing [101]. *P. alni* infects Alder bark via lenticels and/or bark wounds and spreads via secondary growth throughout the plant tissues by medullary rays into the bark xylem [101]. When the pathogen spreads to the bark, damaging effects include the deterioration of phloem tissue and a reduction in mineral transportation. It is unknown whether *P. alni* directly infects Alder leaves and catkins; however, infection of the roots and bark has damaging effects on all compartments of the tree.

Since the 1990s, *P. alni* has been a major issue within many countries and has continued to spread worldwide today, causing a range of damage to the *Alnus* species, which is believed to have emerged due to hybridisation within Europe [104,105,106,107]. The parent species of *P. alni* are *P. uniformis* (diploid) and *P. multiformis* (haploid), with the ploidy level showing that *P. alni* contains half of each parental genome; therefore *P. alni* is a triploid homoploid-type taxon [107]. According to Gibbs (2005), this pathogen was first detected in Southern England and has continued to spread throughout Europe [108,109] (Figure 3). It is hypothesised that the European spread of *P. alni* is due to clonal dispersion, since several mitochondrial haplotype variations have been observed, which may imply that multiple sexual hybridisation incidents generated a hybrid from several clones [107]. *P. alni* ssp. *lat.* was formally named by Brasier et al. (2004) and is now divided into three subspecies: *P. alni* ssp. *alni*, *P. alni* ssp. *uniformis*, and *P. alni* ssp. *multiformis*, considering different variants and hybrids amongst the species, with *P. alni* ssp. *alni* being the most aggressive one [110,111,112]. The *P. alni* species complex was later renamed by Husson et al. (2015) to *P. × alni*, *P. × multiformis, and P. uniformis* [112]. Only *P. alni* ssp. *uniformis* has been identified in North America to date [94,109]. The pathogen has also been identified in Ireland [110]. Symptoms of the *P. alni* complex species include dieback of branches, crown thinning, bleeding bark, dark spotting on stems, stunted leaf growth, premature leaf abscission, leaf yellowing, root rot, necrotic lesions on Alder bark, an excessive number of catkins on the species due to stress, and/or death [4,11]. Alder trees situated around riverbanks and flood plains have greater exposure to infection as the soil-borne pathogen spreads through water, infecting Alder bark and the roots, thus spreading the disease to a greater extent [3,4]. The pathogen spreads via streams, irrigation, flooding events, canals, drainage water, and other pathways via water systems [113]. Alternative pathways may include the plantation of Alder that originated from an affected nursery, human recreational activities, as well as animal activity [23].

Jung et al. (2018) conducted a thorough review of forestry diseases causing decline and suggested that other *Phytophthora* species such as *P. acerina*, *P. cactorum*, *P. chlamydospora*, *P. gonapodyides*, *P. lacustris*, *P. plurivora*, *P. polonica*, *P. pseudocryptogea*, and *P. siskiyouensis* may also be responsible for diseases associated with Alder dieback, although *P. alni* complex species appears to be the main cause of Alder decline [28,94,114,115,116]. For example, O’Hanlon et al. (2020) first reported the presence of *P. lacustris* causing disease within Alder trees in Ireland, indicating that Alder is at risk of dieback due to other *Phytophthora* species [28]. In Ukraine, Matsiakh et al. (2020) analysed rhizosphere soils of declining Alder where *P. lacustris*, *P. plurivora*, and *P. polonica* were detected, which further indicates that these pathogens play a role in Alder dieback [117]. Tkaczyk et al. (2023) identified *P. plurivora* and *P. cactorum* within soil, root, and water samples associated with declining Alder in Slovakia [118]. Bregant et al. (2020) reported the species *P. plurivora*, which was isolated from declining Alder trees within areas of Turkey, Italy, Spain, and Poland [111]. Furthermore, Bregant et al. (2020) revealed that *P. chlamydospora*, *P. gonapodyides*, and *P. siskiyouensis* potentially caused disease within Alder trees in North America. Zamora-Ballesteros et al. (2016) found that *P. plurivora* caused a high mortality rate within inoculated Alder seedlings, indicating that this species is pathogenic to Alder seedlings [119]. Haque and Diez (2015) conducted in vitro pathogenicity tests of Alder leaves, branches, and twigs with *P. × alni*, *P. × multiformis*, and *P. uniformis*, and it was observed that the pathogen caused foliar necrosis, indicating that the roots and root collar may not be the only source of pathogen inoculum [120].

**Figure 3 microorganisms-11-02187-f003:**
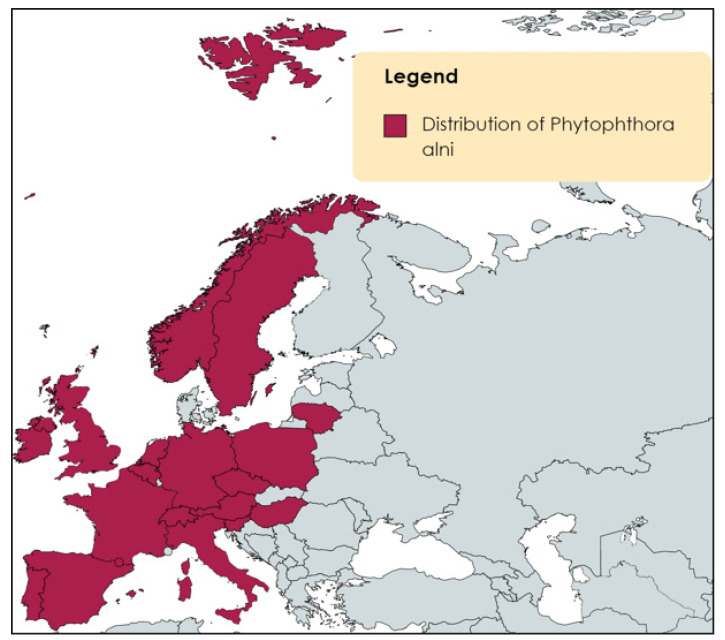
Distribution of *Phytophthora alni* across Europe [109,121].

## 4. The Potential of Microbial Biocontrol Agents in the Fight against Alder Diseases

Biocontrol agents tend to be any beneficial organism, including PGPR and PGPF, which have the ability to provide biological control against harmful pathogenic disease, as well as assisting in acquiring nutrients and amplifying resistive traits within a plant species [33]. Furthermore, the use of PGPR/F can act as a biocontrol agent by enhancing the health of a tree species, thus, a stronger tree can fight/withstand pathogenic infections. Moreover, *A. glutinosa* has many antioxidant, anti-inflammatory, antitumour, and inhibition properties due to the presence of anthraquinones, betulinic acid, diarylheptanoids, flavonoids, phenols, terpenes, tannins, steroids, and saponins; thus, these substances have the potential as BCAs within forestry [17,18,122]. Emerging diseases within forestry are a cause for concern, therefore, the advancement of BCAs within this sector is desirable. BCAs and bio-fertilisers have been used in agriculture for many years for sustainable crop production to minimise the use of harmful pesticides and fertilisers and reduce pathogenic infections, as well as to promote the health and growth of crops [123,124]. This approach is used intermittently within forestry with a few successful applications. Through research, several potential BCAs have been identified within forest ecosystems to reduce pathogenic spread and associated symptoms including dieback and root rot. Utkhede et al. (1997) analysed the antagonistic effects of the bacterium *Enterobacter aerogenes* against the pathogen *P. lateralis*, which tends to affect tree roots within soil and water, causing root rot disease of Lawson cypress trees [125]. D’Souza et al. (2005) investigated several potential BCAs including *Acacia extensa*, *A. stenoptera*, *A. alata*, and *A. pulchella* against the root rot disease caused by *P. cinnamomi*, to observe that these genera have the potential to provide protection to susceptible species [126]. With the *Alnus* species being under threat from the harmful effects of *Phytophthora*, such studies could potentially aid in determining the association between *P. alni* and the *Alnus* species and thus determining potential BCAs to prevent infections.

Several bacterial strains could potentially provide BCAs against emerging tree pathogens [127]. For example, for the pathogen *Hymenoscyphus fraxineus*, a study performed on the microbial community associated with ash tree leaves (*Fraxinus excelsior*) discovered elevated amounts of isolates within the genera *Luteimonas*, *Aureimonas*, *Pseudomonas*, *Bacillus*, and *Paenibacillus* present within tolerant trees, which potentially inhibit the invasion and spread of ash dieback disease (*H. fraxineus*) by direct competition or inducing systemic resistance, as well as the potential production of antagonistic metabolites and/or antifungal compounds [128,129]. Similarly, Becker et al. (2022) explored the microbiome of tolerant ash trees, and it was determined that the bacterium *Aureimonas altamirensis* had the ability to reduce *H. fraxineus* infections by niche exclusion and inducing mechanisms such as exopolysaccharides production and protein secretion [130]. The plant growth-promoting members of the genus *Streptomyces* have proven to be a prime choice as BCAs since these are readily available in nature and have the capacity to control pathogenic infections [9,131,132,133]. Liu et al. (2009) discovered that the BCA metabolite *daidzein* was produced by the bacterial genus *Streptomyces* and isolated it from a root of *A. nepalensis* alongside the *Streptomyces* bacterium, which may have potential antioxidant, anti-inflammatory, and anticancer effects [134,135]. The genus *Streptomyces* can produce the enzyme chitinase, which has been shown to have antagonistic effects against fungi due to the ability to degrade chitin present in the cell walls of fungi as well as reduce spore germination [136].

Biocontrol modes of action can be direct (interaction between pathogens and BCAs) or indirect (interaction between BCAs and plants to enhance their health and resistance to (a)biotic stressors) [137]. Direct biocontrol factors include antimicrobial metabolites, hydrolytic enzymes, quorum sensing, quenching, and resource competition, as well as siderophores. Indirect biocontrol factors include organisms triggering enhanced immunity for plants, environmental adaption, and hormone modulation. Microorganisms have the ability to produce microbe-associated molecular patterns (MAMPs) or damage-associated molecular patterns (DAMPs) like flagellin, glucan, xylan, or chitin [56]. These patterns are recognised by pattern-recognition receptors (PRRs) or other elicitors such as volatile organic compounds (VOCs) or siderophores, which are detected by specific receptors. Activation of these receptors initiates various signalling mechanisms that serve as precursors for the production of phytohormones, triggering defensive pathways. The kinase pathway can phosphorylate transcription factors that regulate the expression of early and late response genes [56]. There are many genes that have an association with biocontrol agents and defence-related genes [138]. Each BCA has thousands of genes present within its genome (depending on its genome size); therefore, genes with biocontrol traits can differ from one species to another. Nelkner et al. (2019) used transcription and genome mining in order to identify genes that are related to biocontrol traits and biosynthetic genes within a *Pseudomonas* strain [139]. In this study, biocontrol activity included the production of siderophores, secondary metabolites, and antibiotics (2,4-Diacetylphloroglucinol (DAPG), hydrogen cyanide (HCN) synthesis, pathogen inhibition genes (iron acquisition, exoprotease activity, and chitinase activity), resistance genes (PRRs), exopolysaccharides genes, lipopolysaccharides genes, metabolism genes, detoxification genes, and genes related to ISR and growth-promoting compounds (VOCs, acetoin, 2,3-butanediol, growth regulators/plant hormones, and phosphate solubilisation) [139]. Genes also involved in plant defence include the phenylpropanoid pathway gene-expressing phenylalanine ammonia-lyase (PAL), which facilitates the deamination process when phenylalanine is converted into cinnamate and ammonia, as well as the lipoxygenase pathway gene encoding hydroxyperoxide lyase (HPL) [56]. Mitogen-activated protein kinase (MAPK)- and cyclic adenosine monophosphate (cAMP)-associated molecules represent widespread intracellular signalling pathways that integrate extracellular stimuli, alter the expression and functionality of receptors, and regulate processes such as cell survival and neuroplasticity [140]. TGA transcription factors are vital regulators of diverse cellular processes which link to hormonal pathways, interacting proteins, and regulatory elements [141]. In addition, the phytohormones brassinosteroids (BRs) and jasmonates (JAs) also aid in the regulation of plant growth, development, and defensive response [142].

Chitinase has been used in agriculture during crop cultivation for disease management, improved growth, and greater yields [143]. Kumar et al. (2018) highlighted various ways in which chitinase gene expression has been used in agriculture to combat fungal diseases [143]. Auxins such as IAA and phenylacetic acid (PAA) are well-known bio-agents within agriculture due to their plant growth-promoting abilities and provide potential control against *Fusarium* pathogens, which have been found within soils, causing root rot [144,145]. IAA has been abundantly produced from PGPR isolates associated with forest trees; therefore they may have the potential for BCAs within this sector since they are proven to be advantageous in agriculture [38]. Trinh et al. (2022) discovered that the rhizobacteria *Bacillus subtilis*, isolated from the rhizosphere roots of black pepper, has potential as a bio-agent for *Fusarium* antagonism [144]. Fungal endophytes could potentially provide BCAs against emerging tree pathogens [146]. For example, Kosawang et al. (2017) performed in vitro antagonistic assays of *Sclerostagonospora* sp., *Setomelanomma holmii*, *Epicoccum nigrum*, *Boeremia exigua*, and *Fusarium* sp. against the dieback-causing fungus, *H. fraxineus*, and it was observed that these endophytes are potential BCAs [147]. Halecker et al. (2020) analysed the antagonistic effects of the fungal endophyte *Hypoxylon rubiginosum* on *H. fraxineus*, and it was determined that this endophyte has potential as a BCA due to the production of metabolites that are toxic to the pathogen [128,148]. This investigative approach is beneficial to determine potential pathogen-fighting microbes present within tolerant tree species that can aid in the protection of trees that are susceptible to certain diseases. See Table 1 for a summary of BCAs.

The most commonly used biocontrol agents belong to the genera *Pseudomonas*, *Bacillus*, and *Trichoderma* [29]. For example, *Pseudomonas* (*P. pudida* 06909) was used as a biocontrol agent against *Phytophthora* root rot within citrus orchids [149], *P. chlororaphis* has antagonistic effects against the cacao rot pathogen *Ph. Palmivora* [150], and *Bacillus amyloliquefaciens* isolated from ginseng rhizosphere induced systemic resistance to *Ph. cactorum* [151]. In addition, *Trichoderma virens*, *Trichoderma harzianum*, *Trichoderma asperellum*, and *Trichoderma spirale* isolated from cocoa show antagonistic effects against *Ph. Palmivora* [152], and *Trichoderma saturnisporum* also showed antagonistic effects against the *Phytophthora* spp. [29,153]. Abbas et al. (2022) conducted an extensive review regarding the identification of multiple biocontrol encoding genes within the *Trichoderma* spp. against *R. solani* including secondary metabolite genes, siderophores genes, signalling molecular genes, cell wall degradation enzymatic genes, and plant growth regulatory genes [154]. Furthermore, G-protein coupled receptor (GPCR) genes, adenylate cyclase genes, protein kinase-A genes, and transcription factor proteins are important genes found in *Trichoderma* for biocontrol against *R. solani* [154]. Moreover, *Trichoderma* has the ability to generate chitinase, which can minimise the survival of *R. solani* through the activation of the expression of chitinase genes [155]. Hernando José et al. (2021) discuss several isolated endophytic bacteria, fungi, and metabolites that have shown antagonistic and biocontrol activities as well as induced mechanisms against *Phytophthora* pathogens [156]. Some bacterial examples include *Pseudomonas fluorescence* isolated from the flowering vine saw greenbrier, which showed antagonistic effects against *P. parasitica*, *P. cinnamomi* and *P. palmivora*, as well as *Burkholderia* spp. isolated from the herb huperzine showing the ability to inhibit the growth of *P. capsici*. The isolate *Acinebacter calcoaceticus* from soybeans showed antagonistic effects against *P. sojae*. An isolate from a tomato plant (*Bacillus cereus*) helped to reduce the infection severity of *P. capsici* associated with cacao trees. Several strains of bacteria isolated from cucurbits including *Bacillus*, *Cronobacer*, *Enterobacteriaceae*, *Lactococcus*, *Pantoea*, and *Pediococus* showed antagonistic activity against *P. capsici* using various biocontrol mechanisms. Some fungal examples include species of *Trichoderma*, *Pestalotiopsis*, and *Fusarium* which were isolated from cacao trees and illustrated antagonism against *P. palmivora*. The fungus *Muscodor crispans* was isolated from a pineapple plant and showed inhibition properties against *P. cinnamomi* and *P. palmivora*. Hernando José et al. (2021) discuss numerous other bacterial and fungal strains that showed biocontrol activities and mechanisms against *P. capsici*, *P. infestans*, *P. citricola*, *P. cactorum*, and *P. pini* [156]. It is evident that there are various studies associated with the biocontrol of several *Phytophthora* species, but there are minimal studies regarding *P. alni*.

### 4.1. Improving Plant Resistance via Microbe Inoculation and Genetic Resistant Breeding

In recent studies, a greater focus has been placed on ways to detect and diagnose tree diseases, understand the interactions between trees and pathogens, develop disease-resistant trees, and ultimately optimise soil and tree microbiomes to improve plant health and behaviour [157,158,159]. The microbial communities associated with forest trees provide vital information relative to species health since numerous beneficial microbes have the ability to increase growth, development, productivity, and ecosystem function, as well as improve soil structure, and provide some resistance to pests and pathogenic diseases [30,33]. Alder trees may be inoculated with beneficial microbes in order to improve disease tolerance and environmental stressors, as well as enhance growth and quality. For example, Chandelier et al. (2015) discuss several inoculation methods in order to screen *A. glutinosa* resistance to *P. alni* [160]. These included wound inoculation, stem inoculation, seed and seedling inoculation, inoculation by flooding of root cuttings, and zoospore suspension inoculation. It was determined that the combination of zoospore suspension with flooding root systems was the most reliable since unwounded trees better mimicked natural environmental conditions [70]. Zaspel et al. (2014) investigated a way to increase the resistance of Common Alder against *P. alni* using in vitro and in planta analysis with a cyclolipopeptide (CLP)-producing *Pseudomonas veronii* metabolite (PAZ1) [70]. It was observed that inoculation with PAZ1 showed some inhibition effects as well as growth-promoting effects on the Alder species.

In order to aid the protection and integrity of forestry, natural genetic resistance breeding programmes of several species against biotic stressors have been studied [161]. Examples of resistance programmes include white pine and white pine blister rust resistance, Port Orford cedar and *P. lateralis*, Sitka spruce and white pine weevil resistance, Loblolly pine and fusiform rust resistance, *Pinus radiata* and *Dothistroma pini* resistance, Koa and koa wilt resistance, American beech and beech bark disease resistance, and Dutch elm disease [161,162,163]. Wei and Jousset (2017) suggest an alternative foundation to reach economically novel phenotypes by altering genetic information coupled with plant-associated microbiota [33,164]. Since pathogenic organisms have caused significant economic losses within forestry, greater research and screening techniques are required in order to expand bio-control procedures against forestry pathogens. Marco et al. (2022) extensively review microbe-assisted breeding programmes to date, particularly those associated with crops, and how this has improved the agricultural sector [165]. The Institute of Forest Genetics in Waldsieversdorf, Germany, was working on a selective breeding programme to improve the resistance of *A. glutinosa* to *P. alni*; however, this project appears to be complete since 2018 [166]. A greater focus on the development of natural genetic resistance breeding programmes specifically focused on the *Alnus* genus coupled with identifying species-specific biocontrol agents to minimise the infection of the pathogen *P. alni* will greatly aid in the protection of the *Alnus* species.

**Table 1 microorganisms-11-02187-t001:** Bio-control agents (BCAs) and the plant diseases they suppress via associated biocontrol mechanisms.

Potential BCA	Classification	Pathogens They Potentially Control	Mechanism Expressed Showing Bio-Control Abilities	In Vitro/In Planta/In Vivo/Field/ Commercial	Reference
*A. extensa*, *A. stenoptera*, *A. alata*, *A. pulchella*	Legume (*Acacia*)	*Phytophthora cinnamomi*	Suppression and containment of *Phytophthora cinnamomi* inoculum in soil-infested areas to protect susceptible species and enhance species diversity	In planta	[125]
*Bacillus amyloliquefaciens*	Firmicutes	*Phytophthora cactorum*	Induces systemic resistance	In vitro	[151]
*Pseudomonas chlororaphis*	Gammaproteobacteria	*Phytophthora palmivor*	Antibiosis and suppression; reduce symptom severity; production of lytic enzymes, siderophores, and bio-surfactants	In vitro/in planta	[150]
*Pseudomonas putida 06909*	Gammaproteobacteria	*Phytophthora* spp.	Hypovirulence and suppression; colonise pathogenic hyphae and reduce pathogenic populations	Field	[149]
*Enterobacter aerogenes*	Proteobacteria	*Phytophthora lateralis*	Suppression; reduce disease symptoms and promote plant growth	Field	[124]
*Hypoxylon rubiginosum*	Ascomycetes	*Hymenoscyphus fraxineus*	Antibiosis; production of metabolites that are toxic to *H. fraxineus*	In vitro/in planta	[148]
*Aureimonas altamirensis*	Alphaproteobacteria	*Hymenoscyphus fraxineus*	Colonisation resistance; exopolysaccharide production; protein secretion; stress adaptation genes	In vitro/in planta	[129]
*Bacillus subtilis*	Firmicutes	*Fusarium* sp.	Antibiosis and antagonistic effects	In vitro	[144]
*Pseudomonas fluorescens*	Pseudomonas	*Fusarium oxysporum*, *Rhizoctonia solani*, *Macrophomira phaseolina*, *Fusarium* sp.	Antibiosis, hypovirulence, and suppression; inhibit plant disease, minimise fungal infections, protect seeds and roots from infection; production of secondary metabolites	In vitro/field	[51,52,167,168]
*daidzein*	Isoflavones	No specific pathogen	Antibiosis; antioxidant, anti-inflammatory, and anticancer effects	In vitro/in vivo	[133,134]
*Streptomyces*	Actinobacteria	*Acidovorax* spp., *Fusarium* sp., *Ralstonia solanacearum*, *Xanthomonas* spp., *Sclerotinia sclerotiorum*, *Erwinia amylovora*	Antibiosis and induced resistance; source of bioactive compounds like antimetabolites, antibiotics, extracellular enzymes, and antitumor agents	In vitro/in vivo	[48,130,131,132,133]
Chitinase	Enzyme	*Phytophthora cinnamomi*	Suppression; reduced spread, reduced sporangia production, applied in agriculture during crop cultivation for disease management, improved growth, and greater yields	Commercial/in vitro	[142,143]
IAA	Auxin Plant Hormone	*Fusarium* sp.	Systemic resistance; regulate growth and development processes	Commercial/in vitro	[38,53,145]

### 4.2. Microbiome Engineering and Its Potential Phytophtoria Control

The practice of microbiome engineering has been applied recently to enhance human well-being, agricultural efficiency and combat challenges of bioremediation and contamination within the environment [169,170]. This involves various methods for the modification of host-associated microbiota to improve its health, resilience, and disease tolerance, as well as to benefit surrounding ecosystems [170]. These methods include enrichment, artificial selection, direct evolution, population control, pairwise interactions, microbiome transfer, synthetic microbes, and engineered interactions which are used to manipulate a microbiome to obtain desired characteristics [169,170]. In agriculture, a method used to manage plant diseases involves root microbiome transfer by combining fertile disease-suppressive soils with less fertile disease-conductive soils [171]. For example, Mendes et al. (2011) showed that by mixing these soil types, sugar beet soil was suppressive to the pathogen *R. solani* due to the presence of several species of Proteobacteria, Firmicutes, and Actinobacteria, which play a role in disease suppression. Santhanam et al. (2015) showed that synthetic root-associated microbiota transplants using a mixture of native bacterial isolates (species from *Arthrobacter*, *Bacillus*, and *Pseudomonas*) were effective at reducing the wilt disease of tobacco *Nicotiana attenuate* and that they increased crop resilience [172]. Mukherjee et al. (2022) used a bio-inoculant containing two bacterial strains isolated from chickpeas, *Enterobacter hormaechei* and *Brevundimonas naejangsanensis*, to enhance the productivity of inoculated chickpeas seeds, and it was observed that the consortium increased plant-growth attributes, yields, nutritional content, levels of IAA, siderophore, ammonia, phosphate solubilisation, and potassium solubilisation, as well as antagonistic activity against *Fusarium* sp. due to successful manipulation of the plant microbiome [173]. Wicaksono et al. (2017) inoculated wounds of kiwi fruit plants with *Pseudomonas* strains isolated from the medicinal plant Mānuka, and it was determined that the bacterial strains aided the pathogenic resistance of *P. syringae*, indicating that microbe transfer can be successful in reducing disease severity [174]. There is a great potential for the use of microbiome engineering in forestry as it offers an innovative solution to address various challenges in sustainable forest management. The manipulation of tree-associated microbiota can potentially enhance tree growth and pathogenic resistance as well as optimise nutrient cycles and improve tree tolerance to environmental stressors.

## 5. Conclusions and Future Perspectives

The Common Alder plays a significant ecological role and has many commercial uses; however, the native species is under threat of decline due to the known causal agent *Phytophthora alni*. Other *Phytophthora* species may also contribute to Alder decline, as well as other pests and pathogens to a lesser extent. Understanding the beneficial and pathogenic microbiota associated with Alder is crucial for developing biological solutions to control these threats. Forest trees have many relationships with a wide variety of microbial organisms, which are essential for tree health, nutrient conditions, and ecosystem functionality. Beneficial microbiota, such as PGPR/F, are present in the tree’s endosphere, phyllosphere, and rhizosphere, providing benefits such as nutrient fixation, hormone production, pathogen suppression, and improved plant growth. Research associated with the microbiota of *Alnus* species is limited and mainly focuses on the root and rhizosphere soil. Bacterial phyla such as Actinobacteria, Acidobacteria, Proteobacteria, Bacteroidetes, and Firmicutes, as well as fungal phyla including Ascomycota, Basidiomycota, Ectomycorrhizal fungi, Glomeromycota, and Zygomycota, have been identified in association with the roots and rhizosphere of Alder trees.

Although, there is limited information available on the microbial communities in other parts of Alder trees, such as the leaves, bark, and catkins. Further research is needed to explore the microbial communities within different compartments of Alder trees and their potential interactions. Understanding the core microbiome of Alder and its functional roles will help improve our knowledge of Alder tree health, growth, and tolerance to (a)biotic stressors. Additionally, exploring the use of plant growth-promoting microbes in the forestry sector may result in beneficial applications for sustainable tree production and protection against pathogens. A greater global focus is needed regarding the breeding of microbe-optimised Alder trees that have a degree of resistance to infection caused by *P. alni.* Biocontrol agents have the ability to induce mechanisms to suppress pathogenic infections by direct attack on the pathogen, competition for resources, and indirectly due to a systemic stress response resulting in improved defence and resistance. The identification of species-specific biocontrol agents will aid in the protection of Alder trees from this disease threat. Also, the application of PGPR and PGPF to Alder affected by this disease could potentially pave the way to control its spread. The identification of biocontrol agents which can persist in sufficiently large populations within the microbial communities associated with Alder, and are capable of effectively expressing their biocontrol genes, is a very promising approach to minimise the severity and spread of devastating diseases of Alder and other important tree species. Another current knowledge gap that exists within this sector includes a detailed understanding of the systemic Alder microbiome and its seasonal and geographical dynamics. It is important to understand the different varieties and species of Alder as well as the type of microbes present in tolerant versus susceptible genotypes that can be exploited to improve the health of future Alder stands. Furthermore, having the ability to identify effective *Phytophthora* BCAs that can systematically colonise the tree and persist for long periods of time will aid the long-term survival of Alder to (a)biotic stressors. However, greater knowledge is required to identify effective methods of BCA inoculation of new and existing Alder stands, as well as what positive/negative effects these BCAs have on non-target microbial species within Alder stands.

## Figures and Tables

**Figure 1 microorganisms-11-02187-f001:**
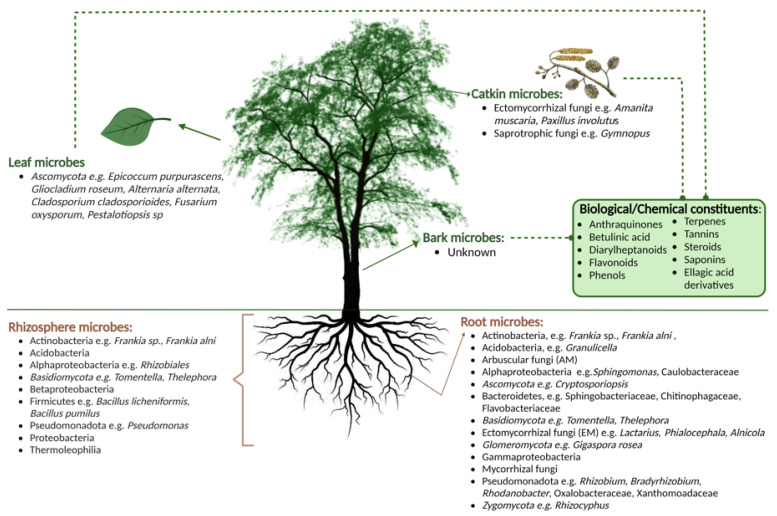
Overview of microbial communities present in different compartments of the Alnus species based on published literature (Created with BioRender.com, accessed on 1 February 2023).

**Figure 2 microorganisms-11-02187-f002:**
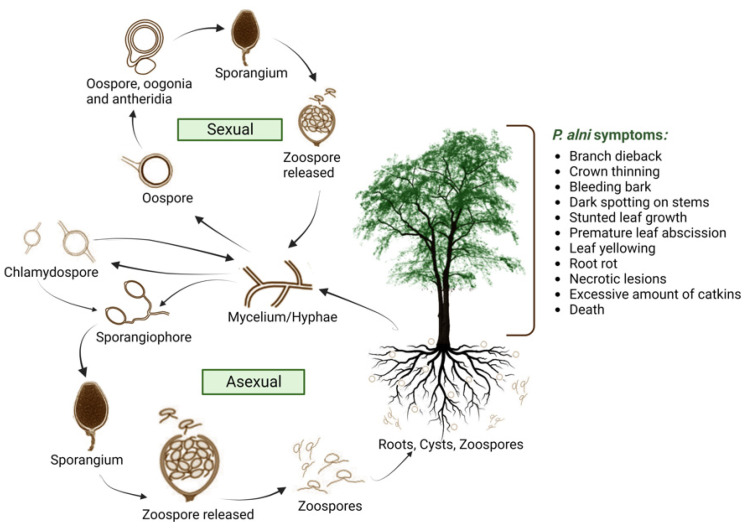
Life cycle of *Phytophthora* showing sexual and asexual phases and associated symptoms on an Alder tree. (Adapted from [97,102] and created with BioRender.com accessed on 17 June 2023).

## Data Availability

Not applicable.

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
