# Peer review of "The Good, the Bad, and the Useable Microbes within the Common Alder (Alnus glutinosa) Microbiome—Potential Bio-Agents to Combat Alder Dieback"

_microorganisms, 2023, doi:10.3390/microorganisms11092187_

Round 1

Reviewer 1 Report

The manuscript is interesting; however, the authors should consider making some improvements before being accepted. Sometimes there are excessively long paragraphs in the text, and the absence of a detailed methodology section makes the manuscript not seem robust. To improve the scientific review paper on "An Overview of Beneficial and Pathogenic Microbiota Associated with Common Alder (Alnus glutinosa)," consider implementing the following recommendations, suggestions, and tips:

§  Clarity and Structure:

Ensure that the paper has a clear and well-defined structure, with appropriate sections such as Introduction, Methods, Results, Discussion, and Conclusion. This will enhance the readability and flow of the review.

Provide a concise and informative abstract that summarizes the main findings and contributions of the review.

§  In-Depth Literature Review:

Conduct a thorough literature search to include the most relevant and up-to-date studies on the topic. Ensure that the review covers a comprehensive range of research papers, including both recent and seminal works.

Consider incorporating meta-analyses, systematic reviews, and other high-quality review papers to provide a well-rounded analysis of the subject matter.

§  Methodology

In a scientific review paper, the following methodological aspects should be included to ensure the review's credibility and reproducibility:

1.      Literature Search Strategy:

Describe the comprehensive literature search strategy used to collect relevant studies and research articles. Specify the databases, keywords, and inclusion/exclusion criteria employed to identify appropriate sources.

2.      Inclusion Criteria:

Clearly outline the criteria used to select studies for the review. These may include the publication date, language, study design, geographical location, and relevance to the topic.

3.      Data Extraction:

Detail the process of data extraction from selected studies. Mention the key information extracted, such as microbial species, their functional roles, interactions with Alnus glutinosa, and any associated ecological or economic impacts.

4.      Quality Assessment:

Discuss the approach taken to assess the quality and reliability of the selected studies. This may involve using standardized tools for quality appraisal or evaluating study methodology and potential biases.

5.      Data Synthesis and Analysis:

Describe the methodology used for synthesizing and analyzing the data obtained from various studies. This may involve narrative synthesis, meta-analysis, or other statistical methods, depending on the nature and availability of data.

6.      Categorization of Microbiota:

Clearly explain the criteria and methods used to categorize the identified microbiota associated with Common Alder. Group them into beneficial and pathogenic categories and elaborate on their respective roles and interactions.

7.      Mapping Microbiota Interactions:

If applicable, describe any network analysis or mapping techniques used to visualize and understand the interactions between Common Alder and its associated microbiota. These visual representations can enhance the reader's comprehension of complex relationships.

8.      Identification of Knowledge Gaps:

Outline the process of identifying knowledge gaps in the existing literature. Mention the criteria used to determine areas where further research is needed and highlight the importance of addressing these gaps.

9.      Strengths and Limitations of the Review:

Acknowledge and discuss the strengths and limitations of the review process. Address any potential biases, shortcomings in data availability, and other factors that may impact the review's findings.

10.   Reproducibility and Transparency:

Emphasize the transparency of the review methodology and ensure that sufficient details are provided to allow other researchers to reproduce the study if needed.

By incorporating these methodological aspects, the scientific review paper will be well-structured, transparent, and robust, enhancing its credibility and contributing to the advancement of knowledge in the field of microbiota associated with Common Alder (Alnus glutinosa).

Author Response

We are greatly thankful to the reviewers for their valuable time, comments and suggestions on the manuscript. We have responded to their comments below.

Response to Reviewer 1 Comments
Point 1: The manuscript is interesting; however, the authors should consider making some improvements before being accepted. Sometimes there are excessively long paragraphs in the text, and the absence of a detailed methodology section makes the manuscript not seem robust. To improve the scientific review paper on "An Overview of Beneficial and Pathogenic Microbiota Associated with Common Alder (Alnus glutinosa)," consider implementing the following recommendations, suggestions, and tips.

Response: Thank you for taking the time to review our manuscript titled "An Overview of Beneficial and Pathogenic Microbiota Associated with Common Alder (Alnus glutinosa)." We appreciate your thoughtful comments and suggestions for improvement. Your feedback is valuable to us in enhancing the quality of our work. The submitted manuscript is a literature review on the available information detailing the bacterial communities, microbial diseases (with a specific focus on Phytophthora) and potential of biocontrol agents to control such diseases in Alder. Little research has been carried out investigating the microbiome of Alder and we believe that collating the available information into one review article will be of genuine interest to readers and researchers in this area. We have carefully reviewed the manuscript and break down lengthy paragraphs into smaller, more digestible sections.

The journals instructions for authors outlines two types of review manuscripts: (1) A review, and (2) A systematic review (this type requires a detailed methodology to be provided in the manuscript). Our submitted manuscript is type 1; therefore, we request to consider our manuscript under “A review” type, which does not require the methodology and sections suggested in the comments. The journal has accepted review articles in similar format recently.  

Moreover, as per suggestion from both reviewers, we have made some significant changes to the overall manuscript to make it more suitable for the journal and attractive to the readers. These additional information/changes are highlighted in yellow within the revised manuscript..

Once again, thank you for your time and expertise in reviewing our work. We look forward to your comments on this revised and improved version of the manuscript.

Reviewer 2 Report

Common Alder is important for ecological system and has many industrial and medical values. The manuscript reviews the known microbiome of Alder, pathogen threats of Alder, beneficial microorganisms, and biological solutions used to control the pathogens. Several comments are listed as follows.

 1.      The title of the manuscript is “An Overview of Beneficial and Pathogenic Microbiota Associated with Common Alder (Alnus glutinosa)”, while the section 3.2 Biocontrol agents (BCA’s) is mainly about trees and plants but not alder.

2.      In the Abstract section, line 20-22, The sentence “This review is focused on collating the literature that has been published around the Alder, threats to the alder tree species, its associated microbial composition and understand their interactions.” should be checked and revised to make the meaning clearer to understand.

 3.      In the section 1 Introduction, line 105-106, the sentence “In particular, the relationship between the pathogenic Phytophthora species and Alnus genus.” Is not complete. It should be revised.

4.      In the section 2. Pathogenic threat, paragraph 4 (from line 180), the name for the Phytophthora species should be written in italic.

5.      In the section 3.1, line 359-360, the sentence “See Error! Reference source not found. for a summary of the microbes identified in the different compartments of the Alnus species.” should be revised.

6.      In the section 3.2, line 458, the sentence “See Error! Reference source not found. for a summary of BCA’s.” should be revised.

7.      For the References, page numbers of some references are missing. Please check and revise according to the reference rules of the journal.

The manuscript is well written, and much information is provided.

Author Response

We are greatly thankful to the reviewers for their valuable time, comments and suggestions on the manuscript. We have responded to their comments below.

Response to Reviewer 2 Comments

Overall comment: Common Alder is important for ecological system and has many industrial and medical values. The manuscript reviews the known microbiome of Alder, pathogen threats of Alder, beneficial microorganisms, and biological solutions used to control the pathogens. Several comments are listed as follows.

Response: We greatly appreciate your insightful comment regarding the significance of Common Alder in ecological systems and its diverse industrial and medical values. Your observation aligns with the core motivation behind our manuscript, "An Overview of Beneficial and Pathogenic Microbiota Associated with Common Alder (Alnus glutinosa)."

Point 1: The title of the manuscript is “An Overview of Beneficial and Pathogenic Microbiota Associated with Common Alder (Alnus glutinosa)”, while the section 3.2 Biocontrol agents (BCA’s) is mainly about trees and plants but not alder. Page 1.

Response: The title of the manuscript has been changed to better reflect the content of the manuscript.

Point 2: In the Abstract section, line 20-22, the sentence “This review is focused on collating the literature that has been published around the Alder, threats to the alder tree species, its associated microbial composition and understand their interactions.” should be checked and revised to make the meaning clearer to understand. Page 1.

Response: The sentence has been checked and revised to make it clear to understand (Now line 21-24 in abstract, page 1).

Point 3: In the section 1 Introduction, line 108-109, the sentence “In particular, the relationship between the pathogenic Phytophthora species and Alnus genus.” Is not complete. It should be revised. Page 3.

Response: This sentence has been removed as it is an irrelevant statement and not necessary to include.

Point 4: In the section 2. Pathogenic threat, paragraph 4 (from line 180), the name for the Phytophthora species should be written in italic. Page 5.

Response: All Phytophthora species were formatted to italics. Please note: This paragraph can now be found on page 9, section 3 – Microbial pathogens of Alder species, paragraph 5. Some sections were re-arranged to provide better flow and specificity to the document.

Point 5: In the section 3.1, line 359-360, the sentence “See Error! Reference source not found. for a summary of the microbes identified in the different compartments of the Alnus species.” should be revised. Page 9.

Response: This error was revised to the appropriate level as ‘see figure 1’. Please note: This paragraph can now be found in section 2.1 on page 5, line 251. Some sections were re-arranged.

Point 6: In the section 3.2, line 458, the sentence “See Error! Reference source not found. for a summary of BCA’s.” should be revised. Page 11.

Response: It has been revised to ‘see table 1’. Please note: This paragraph can now be found in section 4 page 12 (line 514).

Point 7: For the References, page numbers of some references are missing. Please check and revise according to the reference rules of the journal. Pages 17-24.

Response: This comment has been considered to thoroughly check the references. Page numbers were added to all references that were missing.

While revising our manuscript, we have made some additional changes, which are highlighted below for your kind consideration.

  1. Line skipped between each paragraph throughout the text.

Reason: Ensure the text is easier to read and better formatting to provide natural breaks throughout manuscript.

  1. Addition of ‘The presence of’ (line 11) and ‘However’ (line 13) in the Abstract. Page 1.

Reason: Make sentences more readable, clear, and provide a better flow.

  1. Rewording of line 36-37 from ‘Alder dislikes shaded areas and surrounding competitive species’ to ‘Alder preferentially grows in non-shaded areas with little competition’ in the Introduction, paragraph 1. Page 1.

Reason: Sentence needed correction for better clarity of point made.

  1. Removal of ‘However’ from line 50 in Introduction, paragraph 1. Page 2.

Reason: Using word too often in the text.

  1. Addition of word ‘in’ on line 52, Introduction paragraph 1. Page 2.

Reason: Missing word. Grammar correction.

  1. Replace ‘be’ to ‘generate’ and ‘and had less’ to ‘with low’ on line 55 in the Introduction, paragraph 2. Page 2.

Reason: Grammar correction and more appropriate phrasing.

  1. Remove ‘Traditionally, Alder has shown to portray medicinal properties’ on line 60 in the Introduction, paragraph 2. Page 2.

Reason: Irrelevant statement and removing repetition.

  1. Replaced ‘larger’ to ‘an extensive’ on line 71 in the introduction, paragraph 3. Page 2.

Reason: Grammar correction and more appropriate phrasing.

  1. Replaced ‘its importance and promising utilisation has substantially intensified’ to ‘has substantially increased its importance and promising utilisation’ in the Introduction, paragraph 4. Page 2.

Reason: Make the sentence clearer to understand with better phrasing.

  1. Added ‘as a result’ to line 97 in the Introduction, paragraph 4. Page 2.

Reason: Grammar correction/incomplete sentence. Continuing on from previous section.

  1. Added word ‘cultivation’ to line 105 in the Introduction, paragraph 5. Page 3.

Reason: Incomplete sentence/missing word.

  1. Addition of ‘environmentally sustainable’ and ‘to’ to line 111 in the Introduction, paragraph 5. Page 3.

Reason: Improve sentence clarity by adding missing words and better use of English.

  1. Section 2 and 3 were re-arranged. Section 3 was moved up before section 2 and renamed as ‘2. The microbiome of Alder species’ (line 116). Section 2 was moved to after section 3 and renamed as ‘3. Microbial pathogens of Alder species’ (line 286).

Reason: Shifting the focus to Alder and giving the document a better flow.

  1. Inserted the word ‘secure’ in line 129 in section 2, paragraph 1. Page 3.

Reason: Sentence was incomplete. Including this amplifies the point being made here.

  1. Removed word ‘improved’ line 154 in section 2, paragraph 2. Page 4.

Reason: It’s unnecessary to include and makes the sentence confusing. Added the word ‘beneficial’ as it’s more accurate.

  1. Replace ‘like’ with ‘such as’ on line 157 in section 2, paragraph 2. Page 4.

Reason: Grammatical error/better phrasing.

  1. Replace ‘sector’ with ‘topic’ on line 163 in section 2, paragraph 2. Page 4.

Reason: Incorrect language used.

  1. Added words ‘indole acetic acid’, ‘enzymes’, ‘cellulase’, ‘chitinases’ to lines 171/172 in section 2, paragraph 3. Page 4.

Reason: First time they are introduced/abbreviated so it makes sense to mention them at the beginning of the paragraph.

  1. Addition of the phrase ‘to the plant stress hormone’ to line 175 in section 2, paragraph 3. Page 4.

Reason: Further clarify the point as the sentence can be confusing.

  1. Figure 1 was relocated to line 256, between the two paragraphs of section 2.1 ‘what is known about the core microbiome of alder’.

Reason: Better positioning of the image in the section.

  1. Final paragraph in section 3 was moved to be the first paragraph of section 3 (line 287).

Reason: Layout provides a greater focus on Phytophthora and a better flow to the text.

  1. Sentence ‘One of the major pathogens of Alder trees is the plant pathogenic oomycete species Phy-tophthora.’ Added to end of paragraph, section 3, page 7.

Reason: Sentence needed to provide flow to the next paragraph.

  1. Rephrasing of line 315, section 3 paragraph 2.

Reason: Grammar improvement. Sentence was unnecessarily long. Also providing flow from previous section and less repetition.

  1. Figure 2 was moved to line 341 on page 8, between paragraphs 2 and 3 in the section.

Reason: Providing a break in the text and keeping the image with the associated paragraph.

  1. Corrected ‘s.’ to ‘ssp.’ on line 366 in section 3, paragraph 4. Page 9.

Reason: Spelling error/ incorrect terminology previously used.

  1. Section 3.2 was moved and renamed as ‘4.  The potential of microbial biocontrol agents in the fight against Alder diseases’ line 407, page 10.

Reason: Needed to be an independent section.

  1. Replaced ‘in terms of’ with ‘for’ in line 435 in section 4, paragraph 2. Page 11.

Reason: Unnecessary phrasing.

  1. Replaced ‘highlight microbes that prevent against’ with ‘invasion’ on line 438 in section 4, paragraph 2. Page 11.

Reason: Unnecessary use of language, more clarity and sentence reads a lot better.

  1. Replaced ‘colonisation resistance’ with ‘niche exclusion’ on line 443 in section 4, paragraph 2. Page 11.

Reason: Incorrect terminology used and replacement is much clearer.

  1. Replaced ‘has’ to ‘have’ and ‘the genus is’ to ‘these are’, followed by ‘has’ to ‘have’ in lines 445-448 in section 4, paragraph 2. Page 11.

Reason: Grammatical mistake.

  1. Addition of ‘it’ to line 448 in section 4, paragraph 2. Page 11.

Reason: Incorrect Grammar/incomplete sentence.

  1. Section ‘3.4. Main mechanisms and genes involved in biocontrol activities’ was removed from the document (~ 2.5 pages, 1,499 words) as well as the removal of table 2. One paragraph previously in section 3.4 was kept and added to section 4 (lines 455-490).

Reason: Section removed as it was not specific enough to Alder and was focusing more on forestry. Document was also quite long so it needed to be shortened and more specific.

  1. Replaced ‘use’ to ‘used’ and ‘include’ to ‘included’ on lines 471 and 473, section 4 paragraph 3. Page 11.

Reason: Incorrect tense.

  1. Addition of ‘Auxins such as’, ‘and phenylacetic Acetic (PAA) are’ to line 497 in section 4, paragraph 4. Page 12.

Reason: Strengthen point made by including another example of a well-known bio-agent in agriculture.

  1. Replace ‘that have the greatest potential’ to ‘most commonly used’ on line 516 in section 4, paragraph 5. Page 12.

Reason: Sentence reads better and it is a more accurate statement.

  1. Addition of word ‘was’ to line 517 in section 4, paragraph 5. Page 12.

Reason: Missing word.

  1. Replaced ‘inducing’ to ‘induced’ line 520, paragraph 5. Page 12.

Reason: Grammar correction.

  1. Replaced ‘identified’ to ‘show’ line 522, paragraph 5. Page 12.

Reason: Incorrect terminology used.

  1. Replaced ‘Inoculation methods tested’ to ‘these’ on line 563, section 4.1 paragraph 1. Page 13.

Reason: Unnecessary wording and repetition from previous sentence.

  1. Addition of ‘in Waldsieversdorf, Germany’ line 586, section 4.1 paragraph 2. Page 13.

Reason: Providing further information to a generic name.

  1. Renamed table 1 on line 592 from ‘Bio-Control Agents reported against well-known pathogens and their associated biocontrol mechanism.’ To ‘Bio-Control Agents (BCA’s) and the plant diseases they suppress via associated biocontrol mechanisms.’ Page 15.

Reason: Replacement title reads better. ‘Well-known’ was not a good term to use in title.

  1. Converted all references in table 1. Page 15.

Reason: Incorrect format used. E.g. Lee et al. (2015) changed to [151] etc.

  1. Insert references 167 and 168 into the bibliography as they are mentioned in table 1, line 10. Page 15 & 23.

Reason: They were referenced in the table but the full references were accidentally deleted during drafts and editing.

  1. Addition of section ‘3.5 Microbiome engineering and its potential in forestry’ lines 594-624. Page 16.

Reason: Section needed to further strengthen our manuscript as it was incomplete.

  1. Replaced ‘however a major focus relies’ with ‘and mainly focuses on’ line 636 in Conclusions and Future Perspectives, paragraph 1. Page 16.

Reason: Simpler wording and reads well.

  1. Replaced ‘can’ to ‘will’ line 646 in Conclusions and Future Perspectives, paragraph 2. Page 17.

Reason: Wrong tense.

Replace ‘potentially has’ to ‘may result in’ line 648 in Conclusions and Future Perspectives, paragraph 2. Page 17.

Reason: Incorrect phrasing and sentence reads better.

  1. Addition of sentence ‘Another current knowledge gap that exist within this sector includes a detailed under-standing of the systemic Alder microbiome and its seasonal and geographical dynamics. It is important to understand the different varieties and species of Alder as well as the type of microbes present in tolerant versus susceptible genotypes that can be exploited to im-prove the health of future Alder stands. Furthermore, having the ability to identify effective Phytophthora BCAs that can systematically colonise the tree and persist for long periods of time will aid long term survival of alder to (a)biotic stressors. However, greater knowledge is required to identify effective methods of BCA inoculation of new and existing Alder stands, as well as what positive/negative effects these BCAs have on non-target microbial species within Alder stands.’

Reason: Missing sentence needed regarding current knowledge gaps.  

  1. All Latin species names throughout references was formatted to italics. Pages 17-24.

Reason: Incorrect formatting

We sincerely appreciate your valuable feedback and are dedicated to further refining our manuscript to meet the highest standards of scientific rigor and clarity. Thanks a lot.

Round 2

Reviewer 1 Report

The authors have made the corrections, therefore, the manuscript can be accepted